# Chronic Severe Sleep Problems among Non-Nordic Immigrants. Data from a Population Postal Survey in Mid-Sweden

**DOI:** 10.3390/ijerph17217886

**Published:** 2020-10-28

**Authors:** Monica Löfvander, Linda Beckman, Laura von Kobyletzki

**Affiliations:** 1Centre for Clinical Research, Uppsala University, Hospital of Västmanland Västerås, 721 89 Västerås, Sweden; 2Family Medicine and Preventive Medicine Unit, Department of Public Health and Caring Sciences, Uppsala University, Box 564, 751 22 Uppsala, Sweden; 3Department of Public Health Sciences, Karlstad University, 651 88 Karlstad, Sweden; linda.beckman@kau.se; 4Department of Dermatology, Skåne University Hospital, Lund University, 221 00 Malmö, Sweden; laura.von_kobyletzki@med.lu.se

**Keywords:** chronic sleep problems, chronic pain, immigration, population study, public health

## Abstract

Sweden has a large population of both recent and established immigrants with high prevalence of risk factors for ill health. Here, we aimed to explore the prevalence of chronic severe sleep problems (CSSP) among non-Nordic-born persons, and to evaluate the risk for CSSP when fully adjusted for covariates. Our additional hypothesis was that lengthier time since immigration would reduce the risk for CSSP. We used data from a large-population postal survey covering life and health issues among inhabitants in mid-Sweden. Relationship between different countries of birth and CSSP was assessed in logistic analyses for more severe and longstanding pain, sex, employment, mental disability, gastrointestinal problems, and length of stay (short, middle time, and up to ten years of stay). Persons of non-Nordic birth reported significantly more often CSSP, regardless of short or long-term stay. Our findings indicate that non-Nordic birth, regardless of residence time and covariates, was an independent and significant predictor for CSSP. The findings may contribute to increasing awareness in healthcare personnel to recognize chronic sleep problems among immigrant patients. Thus, our study might contribute to developing strategies to enhance health for minorities.

## 1. Introduction

Satisfactory sleep quality is fundamental for good mental and physical health [1,2,3]. Insomnia, or sleeplessness, is a disorder on its own, calling for independent clinical attention, and the diagnostic criteria have recently been made more specific by using frequency and duration for defining the condition.

Longstanding sleep problems increase the risk of mental and physical illness, but have daytime consequences as well, such as slower responses to challenging tasks, all leading to raised costs for health care and accidents [4,5,6,7,8]. Problematic sleep is often associated with negative lifestyle pattern, weak social network, and poor societal integration [3,9,10,11,12,13]. The process of acculturation, unemployment, negative prior life experiences, low self-rated health, and limited access to healthcare are other complicating aspects for many immigrants and refugees [14,15,16]. Patients with multiple ill-health problems interfering with good sleep tend, therefore, to be especially common in multicultural primary care settings, and of special interest for their caregivers [17].

Every sixth person is born outside Sweden (foreign-born or non-Swedish), of which one half or more is born in non-European countries, having insufficient education and small financial margins due to insecure employment or low-income jobs, and pain is a common complaint in this citizen group [18]. Data on longstanding bothering sleep problems among immigrants are few, and show varying results [18,19]. Based on data from the National Survey in Sweden 1996, Sundquist et al. found that Bosnian refugee women on a group level had a much higher risk of undefined sleep difficulties than Swedish-born women, and Taloyan et al. found that Kurdish men twice as often reported sleep difficulties, compared with the Swedish-born men [15,20,21]. A Swiss study found that the non-Western recent immigrants also, after many years of stay, reported more early awakening and trouble falling asleep, as compared with non-immigrants [22]. This finding was expected to be due to high levels of emotional distress. However, rather new immigrants in a Swedish county reported no inferior physical or psychological health, quality of life, wellbeing, or social functioning, compared with their age- and sex-matched native-born controls [23]. Furthermore, in adjusted analyses, the established immigrants in Canada reported fewer sleep troubles, as compared with non-immigrants [24].

Because of mentioned differences in findings, we planned another study, based on a large postal survey in Sweden, focusing on severe and longstanding, i.e., more than three months, sleep problems reported by immigrants with more evident cultural distance than Nordic-born immigrants. The Nordic-born populations have, in general, a relation regarding language, tradition, history, and societal organization (Wikipedia), and are less likely to suffer from discrimination, with its negative impact on mental health [25].

In view of the above, we wished to explore the prevalence of the severe longstanding sleep problems (here: chronic severe sleep problems, CSSP) among foreign-born persons, and if their immigrant status and/or the length of stay in Sweden had independent roles for this serious type of sleep problem. 

Our aim here was to explore the data from a large population postal survey conducted in mid-Sweden for the reported prevalence of CSSP by countries of birth, with focus on non-Nordic-born, and subcategories of persons born in Europe and non-Western countries. In addition, we aimed to explore if the number of years since immigration (length of stay) to Sweden would modify this relationship. 

We hypothesized that especially non-Nordic born persons would have significantly higher risks of CSSP, and also when fully adjusted for well-known covariates for sleep problems. Our additional hypothesis was that lengthier time since immigration would reduce the risk for CSSP. 

## 2. Materials and Methods

### 2.1. Study Conduct and Study Population

We based this paper on a cross-sectional analysis of population-based postal survey data.

The postal survey, Life and Health, was distributed in spring (March to May) 2008, to a randomized population-based sample of 68,710 adults age 18 up to 84 years old in the following five counties: Sörmland, Uppsala, Värmland, Västmanland, and Örebro of mid-Sweden. Three subsequent letters reminded all potential participants of the postal survey. The five counties in mid-Sweden include both big cities and smaller communities. The study sample was a randomly chosen sample of the general population with clusters regarding age, sex, and county and city, or parts of the city for larger cities. The survey letter included an information sheet about the study. The participants accepted their approval in the returned survey. 

The distributed questionnaire collected data on diseases and disorders, lifestyle, work, and education. The set of questions in this questionnaire was derived from international validated questionnaires such as the Public Health Questionnaire and the General Health Questionnaire (GHQ12). Similar postal surveys were distributed every four years in the five counties. Equivalent questions were also used in the national health surveys distributed by The Public Health Agency of Sweden [22].

### 2.2. Outcome Variable

Bothering sleep problems was rated by a single item assessing if, and how often, during the last three months, the participants had experienced sleep problems, in four choices. Participants could respond has not been bothered; has been bothered a single time; has been bothered multiple times; or has been bothered almost all the time, during the past three months. For this study, the item was dichotomized as having chronic severe sleep problems (CSSP = sleep problems multiple times, or almost all the time, for three months or more) or not (no CSSP = (no, or occasional, sleep problems). 

### 2.3. Explanatory Variables

The main explanatory variable was country of birth, categorized into the region of birth. 

Nordic countries were Sweden, Norway, Denmark, and Finland. Non-Nordic countries were all countries outside the four Nordic countries. Sub-analyses were performed with the participants categorized as (i) born in Sweden vs. not born in Sweden (non-Swedish); (ii) born in Europe vs. not born in Europe (non-Europeans); and (iii) born in the Western world vs. not born in Europe, the US, or Australia (non-Western). 

### 2.4. Covariates

Pain was measured in the questionnaire by a single item assessing how often, during the last three months, the participants had experienced bothering pain in the shoulders, neck, back, hips, extremities, stomach, or head. Participants could respond never; single time; multiple times; or almost all the time. Bothering pain was summarized into a binary variable—none/single time, and multiple times/almost all the time. Gastrointestinal problems were handled in the same way, and summarized into a binary variable: no/single time and multiple times/almost all the time. Other binary covariates were sex, mental disability, and unemployment. Age was categorized into five categories: 18–34 years, 35–49 years, 50–64 years, 65–79 years, and >80 years. (cf. Table 1).

### 2.5. Statistical Methods

Cross-tabulation was used to describe the study population. Descriptive analyses were performed for CSSP and the region of birth and the confounding variables. Crude analyses were done for the persons born in the Nordic region and the persons born in the non-Nordic countries, for relations between the baseline variables and CSSP. We performed a stepwise forward logistic regression analysis with the most important covariates according to the literature. We also added economic problems and received social welfare, sick leave, self-reported abuse during the last 12 months, anxiety, atopic dermatitis, diabetes, cancer, tiredness, burn-out, sleep apnea, and consumption of sleeping pills. 

Independence of the above relationships was tested by logistic regression, and adjusting them for the confounding variables. 

We also analyzed if sex modified the relationship between CSSP and countries of birth, as well as CSSP and categories of the length of stay in Sweden. Separate adjusted analyses were performed for three categories of length of stay. 

Sensitivity analyses were performed concerning different definitions of immigration: these analyses included CSSP for the Swedish-born as compared with persons born in any other country (non-Swedish), or those born in a country outside Europe (non-European) as compared with persons born in European countries (European), or persons born outside the Western world (non-Western) as compared with immigrants from the Western world (here in Europe, USA, or Australia). 

Fewer than 5% of the subjects had missing data for some of the studied factors. They were excluded from the analyses.

Statistical significance was defined as a *p*-value < 0.05 or a 95% confidence interval (95% CI). 

The STATA analysis program version 14 was used for statistical analysis. 

### 2.6. Ethical Approval

The study followed the Swedish guidelines for studies in social sciences and humanities, and followed the Declaration of Helsinki. According to Swedish regulations, ethical approval by a medical faculty is no longer required for this type of study, with data from anonymous postal surveys where specific persons cannot be identified and are not comprised by the type of research which the Swedish legislations of research ethics have defined as requiring an ethical approval by a Regional Ethical Board (the Ethical Review Act of Sweden 2003:460 and the Personal Data Act). The decision on this study was made accordingly by the ethical committee in Uppsala, 2003, Sweden, EPN 2012/256.

## 3. Results

### 3.1. Participants

In total, 40,674 adults (45.5% males) returned the questionnaire, corresponding to a response rate of 59.2% (mean age 53.8 years, Std 17.92). The majority was born in Sweden or in the other Nordic countries (37,450; 93.7%). Thus, only 6.3% (*n* = 2493) of the participants were of non-Nordic birth. A minority of them, *n* = 1304 (24.2%) had immigrated during the past ten years, and the majority *n* = 4094 (75.8%) more than ten years previously. Slightly more women than men participated in the study (54.5%). Nearly a fifth of all the participants were unemployed. Swedish-born persons and persons from other Nordic countries had higher response rates (range 59–63% vs. 57–63%, respectively) than persons from non-Nordic countries (41–44%). Women had a higher response rate, 63–67%, vs. men, 51–54%. Persons with higher age, higher education, and employment had higher response rates. 

### 3.2. Statistics

Table 1 shows the distribution of the explanatory and the outcome factors in numbers and percentages. The majority, 78.1%, of the participants had no, or only mild, sleep problems (no CSSP), which was especially common among the youngest participants. One-fifth (21.9%) had CSSP, which was significantly more frequent among the participants of non-Nordic birth (28.5% vs. 21.4%). Furthermore, women, unemployed, and participants with bothering pain, gastrointestinal problems, and/or mental disability had significantly higher frequencies of CSSP. To note, the non-Nordic born participants had still a significantly higher frequency of CSSP after 10 years of residence time. 

Table 2 shows that non-Nordic birth predicted CSSP also when adjusted for the female sex, unemployment, bothering pain, gastrointestinal problems, and mental disability. Age did not confound the region of birth/CSSP relationship. Bothering pain was the most important covariate.

Table 3 shows that the adjusted odds ratios for CSSP were increased for all categories of subgroups of persons born outside Sweden. The odds showed insignificant trends with overlapping confidence intervals for different definitions of countries of birth. Persons born in non-European countries had an aOR of 1.33, and persons born outside the Western world had an aOR of 1.37. The category including all persons born outside Sweden had an aOR of 1.21. The covariates for CSSP were similar in all country of birth categories. There was no change in the relationship between country of birth and sleep problems when adding the other potential covariates. 

Table 4 shows that the odds ratios for non-Nordic birth and other covariates as having CSSP remained at similarly increased levels over the years since immigration, with no improvement regarding chronic severe sleep problems for the non-Nordic participants with longer stay in Sweden. 

## 4. Discussion

This population postal survey in Sweden shows that not being born in a Nordic country was an independent and significant predictor for chronic severe sleep problems (CSSP). It also shows that the increased odds for non-Nordic birth, and the covariates, for CSSP remained at similarly increased levels over the years since immigration. 

We believe our findings in this study to be valid and robust. A single question is used to identify insomnia in many surveys. In this population study, there were four alternatives to the question on sleep problems. The rationale here was to study the most severe cases of insomnia. Thus, we defined CSSP as sleep problems multiple times/almost all the time continuously for three months or more, a definition that corresponds with the DSM-5 definition of sleeplessness [11]. The covariates from this population questionnaire are considered as valid and reliable in this context, and used repeatedly in large population studies in Sweden [22]. Furthermore, the Swedish technology assessment institute has expressed a wish of closing some knowledge gaps regarding subjective assessment in self-rated questionnaires by using binary variables instead of multiple choice [22]. We adhered to this advice also, regarding the covariates pain, mental disability, and gastrointestinal problems. 

Higher frequencies of undefined problematic sleep are found among the immigrants in some other study samples [23]. Only small differences were found here between the foreign-born participants, but with trends to higher odds for the participants of non-Western birth. However, immigrants are not a homogenous concept, including both refugees, asylum seekers, students, and workforce labor from outside Sweden. Such detailed information was not accessible in this survey material. Here, we used non-Nordic birth as defining immigrants, since the Nordic countries are quite similar societies in many aspects. This definition can, however, be discussed. Furthermore, our other categories only vaguely characterize cultural and linguistic distance from Sweden, and may be here reflected in the trend towards a higher prevalence of CSSP among the more recent immigrants. In clinical practice, this might mean that long-term severe sleep problems should be born in mind also for persons from neighboring countries or long-term immigrants, and that severe longstanding problematic sleep can be an independent problem, and not always a component of ill-health or social malfunction. 

Our study showed that both severe pain and mental disability independently predicted CSSP more than short-term and long-term non-Nordic birth, and regardless of sex. However, the non-Nordic birth had a statistically significant, albeit rather small, but yet an independent, impact on longstanding severe sleep problems also in the long term. Hypothetically, this could reflect an adaption to attitudes in communicating health problems in the host country, and/or a withstanding negative acculturation effect with disappointment, segregation, prejudices, and undefined ill-health [15,24,25].

Two novel factors appear in our present paper: non-Nordic birth as an independent risk factor for chronic severe sleep problems, with no actual change over time. Otherwise, we found the same correlates for longstanding severe sleep problems as earlier studies, notably older age, female sex, pain, gastrointestinal problems, mental disability, and unemployment [26,27].

Notably, negative socioeconomic factors or higher body mass index seem not to be linked to altered sleep patterns [1]. Both conditions are common among many immigrant groups, and so are unspecified mental disorders when adjusted for socioeconomic factors and knowledge of Swedish [14,28]. Immigrants on a group level seem overall to have elevated risks independently for depression and psychotic disorders when adjusted for socioeconomic factors and knowledge of Swedish [14,29,30,31]. Further, burnout symptoms are common among foreign-born women in Sweden, again not found to be independently related with sleep problems [21]. An important contributing factor for bad sleep is widespread pain, a condition which tends to be common among foreign-born persons [22]. This type of pain may alter the phase of the deep sleep and increase the pain sensation [3,32,33,34].

The participants in this postal survey were slightly better educated than the general population in Sweden, and more women and old people participated [35]. Therefore, it is possible that especially younger men with little education are underrepresented in this study cohort, regardless of their country of birth. The potential selection might bias the results towards the null value. Also limiting the transferability of our results is the low participation rate of more recent immigrants, probably due to poor proficiency in Swedish. This study might, therefore, have underestimated, e.g., their mental health problems. To compare, in a small study, the prevalence of insomnia among recent immigrants from the Middle East was as high as among Swedes suffering from chronic stress [36]. 

One strength of our study is the large population-based sample of adult individuals. Another strength is the four-level scale for self-reported sleep difficulties. Only the most severe level was in focus for our analysis. The reason was that the most evident risk factors for future ill-health are the chronic severe sleep problems. On the other hand, the simple design of our study has several limitations. First, the cross-sectional nature of self-reported data makes it impossible to conclude the causes and the effects. Second, the level and frequency of chronic sleep problems can be overestimated, or be underreported, by individuals or groups of participants. The extent of this remains unclear. Third, the survey did not include any questions regarding nature or details of sleep problems. In addition, it will be impossible to reproduce our study in later postal surveys, due to differences in alternative answers. Therefore, our conclusions remain to what it is—a study of chronic severe sleep problems and immigrant status framed by its time. Despite all these limitations, it should be possible to transfer our results to Western societies with large, new, and older immigrant populations with varying sociocultural backgrounds.

Large longitudinal epidemiological population studies comparing data over time are justified to give sufficient power to identify the country of birth, length of stay, specific ill-health factors, and sleep quality. Further data of interest in the subject of problematic sleep and immigration could be provided by studies using qualitative methodology in different immigrant groups.

## 5. Conclusions

To conclude, not being born in a Nordic country was an independent and significant predictor for chronic severe sleep problems, regardless of the length of stay in Sweden. The findings may contribute to increasing awareness in healthcare to recognize chronic sleep problems among immigrant patients with, or without, other well-known risk factors. Our study might thus contribute to developing strategies to enhance health for minorities.

## Figures and Tables

**Table 1 ijerph-17-07886-t001:** Characteristics of the study population by no/mild sleep problems (no CSSP) and chronic severe sleep problems (CSSP).

	No CSSP *n* (%)	CSSP *n* (%)	*p*-Value
Observations (n)	31,216 (78.1)	8727 (21.9)	
Age (years)			
18–34	6181 (82.8)	1277 (17.1)	<0.001
35–49	6546 (79.6)	1674 (20.4)	
50–64	8022 (74.5)	2749 (25.5)	
65–79	8955 (78.6)	2433 (21.4)	
>80	1512 (71.8)	594 (28.2)	
Region of birth			
Any Nordic country	29,434 (78.6)	8016 (21.4)	<0.001
Non-Nordic countries	1782 (71.5)	711 (28.5)	
Sex			
Men	15,242 (83.7)	2962 (16.3)	<0.001
Women	15,974 (73.5)	5765 (26.5)	
Unemployed			
No	9469 (85.0)	1674 (15.0)	<0.001
Yes	1940 (71.59)	770 (28.4)	
Pain ^#^			
No	15,259 (89.4)	1815 (19.6)	<0.001
Yes	15,556 (69.7)	6778 (30.4)	
Gastrointestinal problems ^#^			
No	26,456 (81.0)	6195 (19.0)	<0.001
Yes	2243 (59.5)	1526 (40.5)	
Mental disability			
No	25,529 (82.7)	5324 (17.3)	<0.001
Yes	3314 (57.3)	2470 (42.7)	
>10 years of residency ^##^			
No	1006 (79.3)	263 (20.7)	<0.001
Yes	2847 (71.3)	1146 (28.7)	

Note: ^#^ Bothering condition for more than 3 months; ^##^ Number based on which year people immigrated to Sweden.

**Table 2 ijerph-17-07886-t002:** Adjusted odds ratios (aOR) for CSSP, with 95% confidence intervals (95% CI).

CSSP	aOR	95% CI
All non-Nordic-born (Ref: Nordic born)	1.34	1.12∓1.61 ***
Woman	1.55	1.40∓1.71 ***
Unemployed	1.84	1.65∓2.06 ***
Pain ^#^	2.60	2.35∓2.89 ***
Gastrointestinal problem ^#^	1.77	1.52∓2.05 ***
Mental disability	1.99	1.73∓2.30 ***

*** *p*-value < 0.001. Note: CSSP = chronic severe sleep problems for more than 3 months. Reference category is all other as opposed to the labelled category. ^#^ Bothering condition for more than 3 months.

**Table 3 ijerph-17-07886-t003:** Adjusted odds ratios (aOR) with 95% confidence intervals for different countries of birth as having CSSP.

	Europe		Western Countries		Sweden	
	aOR	95% CI	aOR	95% CI	aOR	95% CI
CSSP						
Born outside ^1^	1.33	1.04–1.71 **	1.37	1.07–1.76 **	1.21	1.04–1.40 **
Woman	1.55	1.40–1.70 ***	1.55	1.40–1.70 ***	1.55	1.40–1.70 ***
Unemployed	1.85	1.66–2.06 ***	1.85	1.66–2.06 ***	1.85	1.66–2.06 ***
Pain ^#^	2.60	2.35–2.89 ***	2.60	2.35–2.89 ***	2.60	2.35–2.89 ***
Gastrointestinal problem ^#^	1.78	1.52–2.01 ***	1.76	1.52–2.01 ***	1.77	1.52–2.06 ***
Mental disability	1.99	1.75–2.30 ***	1.99	1.72–2.30 ***	1.99	1.72–2.30 ***

** *p*-value < 0.01 *** *p*-value < 0.001 ^1^ Born outside Europe, outside the Western world, or outside Sweden, respectively. ^#^ Reference category is all other as opposed to the labelled category. ^#^ Bothering condition for more than 3 months.

**Table 4 ijerph-17-07886-t004:** Adjusted (Adj.) odds ratios (OR) with 95% confidence intervals (95% CI) as having CSSP, when adjusted for categories of lengths of stay in Sweden.

	Adj. 3 Years		Adj. 5 Years		Adj. 10 Years	
Variables						
Non-Nordic born (Ref: Nordic-born)	1.42 ***	(1.09–1.86)	1.42 ***	(1.09–1.85)	1.44 ***	(1.10–1.87)
Unemployed	1.66 ***	(1.25–2.20)	1.65 ***	(1.24–2.18)	1.67 ***	(1.26–2.21)
Pain ^#^	2.60 ***	(1.97–3.45)	2.61 ***	(1.97–3.45)	2.56 ***	(1.95–3.41)
Woman	1.12	(0.89–1.45)	1.11	(0.85–1.44)	1.12	(0.86–1.45)
Gastrointestinal problems ^#^	2.13 ***	(1.46–3.13)	2.15 ***	(1.47–3.15)	2.16 ***	(1.47–3.16)
Mental disability	2.18 ***	(1.48–3.21)	2.14 ***	(1.45–3.14)	2.14 ***	(1.46–3.14)
Length of stay (years)						
Up to 3	2.05 **	(1.10–3.81)				
Up to 5			1.53	(0.96–2.44)		
Up to 10					1.36 **	(1.00–1.85)
Observations (n)	1613		1613		1613	

** *p*-value < 0.01; *** *p*-value < 0.001. Note: The analysis was adjusted for all variables. Note: CSSP = chronic severe sleep problems more than 3 months. ^#^ Reference category is all other, as opposed to the labelled category. ^#^ Bothering condition for more than 3 months.

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
