# Peer review of "Chronic Severe Sleep Problems among Non-Nordic Immigrants. Data from a Population Postal Survey in Mid-Sweden"

_ijerph, 2020, doi:10.3390/ijerph17217886_

Round 1

Reviewer 1 Report

The authors explored the factors contributing to chronic severe sleep problems (CSSP) in a large population postal survey database.

Major comments:
- A major limitation of the study is the representativeness of the cohort. It is recommended that the authors compared the demographics of the cohort with those of the general population.
- Very few covariates were included into the multivariable models. It is recommended to include more covariates. The authors might then use stepwise variable selection method to compose the final models.
- Throughout the article, some spelling and grammatical errors were noted. Major English editing is recommended.

Minor comments:
- (Line 17) The words "chronic sleep problems" should be "chronic severe sleep problems".
- Please remove the first paragraph (line 32-35).
- (Line 126) "<05" should be "<0.05"
- Please add subheadings to sections 3.1 and 3.2.
- (Line 186) "ucountry" should be "country".

Author Response

We sincerely thank the reviewers very much for their work with our manuscript Chronic Severe Sleep Problems Among Non-Nordic Immigrants We have listened to your suggestions and comments and made our best to answer your questions and rewrite some, or add, some parts.

Reviewer 1.

Comments and Suggestions for Authors

The authors explored the factors contributing to chronic severe sleep problems (CSSP) in a large population postal survey database.

Major comments:

"- A major limitation of the study is the representativeness of the cohort. It is recommended that the authors compared the demographics of the cohort with those of the general population."

Response: Thank you for your comment. We have added texts about representativeness in the method and the discussion sections.

The five counties in mid-Sweden include both big cities and smaller communities. The study sample was a randomly chosen sample of the general population with clusters regarding age, sex and county and city or parts of the city for larger cities. The survey letter included an information sheet about the study. The participants accepted their approval in the returned survey. LL 101-105

The participants in this postal survey were slightly better educated than the general population in Sweden (REF 35). More women and older people participated making it possible that especially men with poor education are underrepresented in this study cohort like in many other large public surveys. In this study, the potential selection might bias the results towards the null value.

LL 294-297

"- Very few covariates were included into the multivariable models. It is recommended to include more covariates. The authors might then use stepwise variable selection method to compose the final models."

Response: Thank you. We have now a new text and added a number of other co-variates that possibly could of interest in this context..See  LL 137-141 and 210-212

We performed a stepwise forward logistic regression analysis with the most important co-variates were chosen according to the literature. We also added economic problems and received social welfare, sick leave, self-reported abuse during the last 12 months, anxiety, atopic dermatitis, diabetes, cancer, tiredness, burn-out, sleep apnea and consumption of sleeping pills.

There was no change in the relationship between country of birth and sleep problems when adding the other potential co-variates.

LL 210-212

"- Throughout the article, some spelling and grammatical errors were noted. Major English editing is recommended."

Response: A language correction will be performed

"Minor comments:

- (Line 17) The words "chronic sleep problems" should be "chronic severe sleep problems".

Response: Thank you, Done LL.21.

"- Please remove the first paragraph (line 32-35)."

Response: Thank you. Done LL 40..........

"- (Line 126) "<05" should be "<0.05" "

Response: Done LL.164.......

"- Please add subheadings to sections 3.1 and 3.2. L 169 and 181

  • (Line 186) "ucountry" should be "country"."

  • Response: Thank you. Done.

Reviewer 2 Report

Thank you for your submission and excellent research. I hope you find my comments useful. First of all, I am not sure what your hypothesis is regarding why non-Nordic country birth would be associated with CSSP. This finding is helpful for clinicians to help identify potential problems, but of interest to me at least is why. What is your hypothesis? I didn't see much discussed about this in your discussion section. I am also not quite sure why you focused on not being born in a Nordic country as your main conclusion, when you found other factors were associated with CSSP as well. I suspect this is because the immigrant status finding was more novel and new, nevertheless, including your other significant findings in the conclusion of your abstract and in the conclusion section of your paper would not take up much space and would making it easier for readers to get the full picture of your analysis. Overall, however, excellent work with interesting findings that likely are meaningful clinically.

Author Response

We sincerely thank the reviewers very much for their work with our manuscript Chronic Severe Sleep Problems Among Non-Nordic Immigrants We have listened to your suggestions and comments and made our best to answer your questions and rewrite some, or add, some parts.

Thank you for your submission and excellent research. I hope you find my comments useful. First of all, I am not sure what your hypothesis is regarding why non-Nordic country birth would be associated with CSSP. This finding is helpful for clinicians to help identify potential problems, but of interest to me at least is why. What is your hypothesis? I didn't see much discussed about this in your discussion section. I am also not quite sure why you focused on not being born in a Nordic country as your main conclusion, when you found other factors were associated with CSSP as well. I suspect this is because the immigrant status finding was more novel and new, nevertheless, including your other significant findings in the conclusion of your abstract and in the conclusion section of your paper would not take up much space and would making it easier for readers to get the full picture of your analysis. Overall, however, excellent work with interesting findings that likely are meaningful clinically.

Response

Thank you for your valuable comments. Based on that we have added more text to motivate our study, and the hypotheses. We also add three more references and rephrased the text in the abstract and the conclusions. Please, see below.

LL 62-75… A Swiss study found that the non-Western recent immigrants also after many years of stay, reported more early awakening and trouble falling asleep, as compared with non-immigrants [22]. This finding was expected to be due to high levels of emotional distress. However, rather new immigrants in a Swedish county reported no inferior physical or psychological health, quality-of-life, well-being or social functioning compared with their age- and sex-matched native-born pairs [23]. Furthermore, in adjusted analyses the established immigrants in Canada reported fewer sleep troubles as compared with non-immigrants [24]. Because of mentioned differences in findings, we planned another study based on a large postal survey in Sweden focusing on severe and longstanding, i.e. more than three months, severe sleep problems reported by immigrants with more evident cultural distance than Nordic-born immigrants. The Nordic-born populations have, in general, a relation regarding language, tradition, history and societal organization (Wikipedia) and are less likely to suffer from discrimination with its negative impact on mental health [25].

Our main hypothesis was that especially non-Nordic born persons would have significantly higher risks of CSSP, also when fully adjusted for the many known covariates for bad sleep, exemplified with longstanding severe conditions such as pain, mental illness, social disadvantages and lifestyle. Our additional hypothesis was that lengthier time since immigration would reduce the risk for CSSP.

We hypothesized that especially non-Nordic born persons would have significantly higher risks of CSSP, also when fully adjusted for well-known covariates for sleep problems. Our additional hypothesis was that lengthier time since immigration would reduce the risk for CSSP.

See LL 85-88 (also reviewer 3)

Rephrased the text in the abstract, summary and the conclusions LL   22-24.LL 232-236………..LL 314-318…

Reviewer 3 Report

This is an interesting study to examine whether foreign-born was an independent factor for chronic severe sleep problems (CSSP) and how the number of years since immigration (length of stay) to Sweden would modify this relationship. Authors carried out a postal survey in Mid-Sweden and they concluded that Non-Nordic birth, regardless of residence time, was an independent and significant predictor for CSSP. I do think the relevant findings may increase awareness to recognize CSSP among immigrant patients and develop strategies to enhance health for minorities. Nevertheless, I still have a few major and other minor concerns.

Major concern

-- Authors should address the study design in Methods. Did this study use the cross-sectional study design?

-- The statements of Methods in this study should be improved. In my opinion, this study recruited the subjects with CSSP as study group and those without CSSP as comparison group. Then, authors detected the differences of characteristics and birth country between CSSP subjects and non-CSSP subjects. Sample selection criteria (CSSP and non-CSSP subjects’ selection) should address in the first part of Methods (Subtitle: Sample Selection). The stages of study procedure should be clear stated in the study. In addition, a better study will have a flow chart which assessed all refers and potential candidates

--How authors deal with the missing data in the returned questionnaires?

--This study identified CSSP only by using simple questionnaire. I considered that this was a limitation in this study. Authors should address the relevant statements in Discussion.

-- In Introduction, authors addressed that “very sixth person is born outside Sweden (foreign-born or non-Swedish) of which one half or more is born in non-European countries having insufficient education and small financial margins due to insecure employment or low-income jobs, and pain is a common complaint in this citizen group [18]”. However, in Discussion, authors stated that “Notably, socioeconomic factors or higher body mass index have not been found linked altered sleep patterns [1]. Immigrants on a group level seem to have elevated risks independently for depression and psychotic disorders when adjusted for socioeconomic factors and knowledge of Swedish [14], [28].” Relevant statements may contribute to misleading. Due to the unclear roles of socioeconomic factors in the connection between birth country and CSSP, authors should provide associated information (about the socioeconomic factors) for the readers and considered these factors in the adjustment models. If this survey did not collect relevant information, I suggested authors to clearly discuss relevant issue in the Discussion.

-- Table 4 shows that the odds ratios for non-Nordic birth and other covariates as having CSSP remained at similarly increased levels over the years since immigration. Authors emphasized that there was a trend with overlapping confidence intervals towards decreasing odds for long-term immigrants with more than 10 years since immigration. How authors carried out relevant analyses? Did authors classify subjects by using length of stay and performed adjusted logistic models in every classification. I’m not sure the meaning of ‘Adj. 3 years’, ‘Adj. 5 years’, and ‘Adj. 10 years’ in Table 4. I recommend authors to improve this Table and provide clear statements in Results.

Minor concern

--The aim of the study should be emphasized.

--More literature review should be addressed in Discussion.

Author Response

We sincerely thank the reviewers very much for their work with our manuscript Chronic Severe Sleep Problems Among Non-Nordic Immigrants We have listened to your suggestions and comments and made our best to answer your questions and rewrite some, or add, some parts.

"This is an interesting study to examine whether foreign-born was an independent factor for chronic severe sleep problems (CSSP) and how the number of years since immigration (length of stay) to Sweden would modify this relationship. Authors carried out a postal survey in Mid-Sweden and they concluded that Non-Nordic birth, regardless of residence time, was an independent and significant predictor for CSSP. I do think the relevant findings may increase awareness to recognize CSSP among immigrant patients and develop strategies to enhance health for minorities. Nevertheless, I still havefew major and other minor concerns."

Reply: Thank you for your valuable comments. Our answers appear below.

Major concern

"-- Authors should address the study design in Methods. Did this study use the cross-sectional study design?"

Reply: This paper is based on a cross-sectional analysis of population-based postal survey data. LL...91-92.

"-- The statements of Methods in this study should be improved. In my opinion, this study recruited the subjects with CSSP as study group and those without CSSP as comparison group. Then, authors detected the differences of characteristics and birth country between CSSP subjects and non-CSSP subjects.Sample selection criteria (CSSP and non-CSSP subjects’ selection) should address in the first part of Methods (Subtitle: Sample Selection). The stages of study procedure should be clear stated in the study. In addition, a better study will have a flow chart which assessed all refers and potential candidates"

Reply: Subheadings are added as suggested LL 91 and on all refers and potential candidates

Three subsequent letters reminded all potential participants of the postal survey. is now added in text L 96. Sorry to say, the statisticians could not provide any flow-chart. Further; The five counties in mid-Sweden include both big cities and smaller communities. The study sample was a randomly chosen sample of the general population with clusters regarding age, sex and county and city or parts of the city for larger cities. The survey letter included an information sheet about the study. The participants accepted their approval in the returned survey. LL 101-105

"--How authors deal with the missing data in the returned questionnaires?"

Fewer than 5% of the subjects had missing data for some of the studied factors. They were excluded from the analyses. LL.162-163.

"--This study identified CSSP only by using simple questionnaire. I considered that this was a limitation in this study. Authors should address the relevant statements in Discussion."

Reply: Thank you for your comment. We have reformulated and added some text. We hope this part is satisfying now. LL 296-317 (also reviewer 1)

The participants in this postal survey were slightly better educated than the general population in Sweden, and more women, and more old people, participated [35]. Therefore, it is possible that especially younger men with little education are underrepresented in this study cohort, regardless of their country of birth. The potential selection might bias the results towards the null value. Also limiting the transferability of our results is the low participation rate of more recent immigrants, probably due to poor proficiency in Swedish. This study might, therefore, have underestimated e.g., their mental health problems. To compare, in a small study the prevalence of insomnia among recent immigrants from Middle East was as high as among Swedes suffering from chronic stress [32].

    One strength of our study is the large population-based sample of adult individuals. Another strength is the four-level scale for self-reported sleep-difficulties. Only the most severe level was in focus for our analysis. The reason was that the most evident risk-factor for future ill-health is the chronic severe sleep problems. On the other hand, the simple design of our study has several limitations. First, the cross-sectional nature of self-reported data makes it impossible to conclude the causes and the effects. Second, the level and frequency of chronic sleep problems can be overestimated, or be underreported, by individuals or groups of participants. The extent of this remains unclear. Third, the survey did not include any questions regarding nature or details of sleep problems. In addition, it will be impossible to reproduce our study in later postal surveys due to differences in alternative answers. Therefore, our conclusions remain to what it is – a study of chronic severe sleep problems and immigrant status framed by its time.

"--In Introduction, authors addressed that “very sixth person is born outside Sweden (foreign-born or non-Swedish) of which one half or more is born in non-European countries having insufficient education and small financial margins due to insecure employment or low-income jobs, and pain is a common complaint in this citizen group [18]”. However, in Discussion, authors stated that “Notably, socioeconomic factors or higher body mass index have not been found linked altered sleep patterns [1]. Immigrants on a group level seem to have elevated risks independently for depression and psychotic disorders when adjusted for socioeconomic factors and knowledge of Swedish [14], [28].” Relevant statements may contribute to misleading. Due to the unclear roles of socioeconomic factors in the connection between birth country and CSSP, authors should provide associated information (about the socioeconomic factors) for the readers and considered these factors in the adjustment models. If this survey did not collect relevant information, I suggested authors to clearly discuss relevant issue in the Discussion."

Reply: –Added text: We performed a stepwise forward logistic regression analysis with the most important co-variates according to the literature. We also added economic problems and received social welfare, sick leave, self-reported abuse during the last 12 months, anxiety, atopic dermatitis, diabetes, cancer, tiredness, burn-out, sleep apnea, consumption of sleeping pills. LL. 146-150. Also slightly revised text Notably, negative socioeconomic factors or higher body mass index seem not to be linked to altered sleep patterns [1]. Both conditions are common among many immigrant groups, and so are unspecified mental disorders when adjusted for socioeconomic factors and knowledge of Swedish [14], [28]. LL 286-289

"-- Table 4 shows that the odds ratios for non-Nordic birth and other covariates as having CSSP remained at similarly increased levels over the years since immigration. Authors emphasized that there was a trend with overlapping confidence intervals towards decreasing odds for long-term immigrants with more than 10 years since immigration. How authors carried out relevant analyses? Did authors classify subjects by using length of stay and performed adjusted logistic models in every classification. I’m not sure the meaning of ‘Adj. 3 years’, ‘Adj. 5 years’, and ‘Adj. 10 years’ in Table 4. I recommend authors to improve this Table and provide clear statements in Results."

Reply: Separate adjusted analysis were performed for three categories of length of stay .LL 154-155. No improvement regarding chronic severe sleep problems for the immigrants with longer stay in Sweden. LL 232-233. Table 4 headings revised L 235

"Minor concern

--The aim of the study should be emphasized."

Reply: Thank you. We have reformulated and added more details

Reformulated aim, partly  L 87

See added hypothesis LL.92-105

"--More literature review should be addressed in Discussion"

Reply:

We have added three references both in the Introduction…22,24,25for the sake of the context and background LL 69-82 (also reviewer 2).

and in the Discussion L297-298 ref no 35. (shared with reviewer 1)

The participants in this postal survey were slightly better educated than the general population in Sweden, and more women, and more old people, participated [35]. Therefore, it is possible that especially younger men with little education are underrepresented in this study cohort, regardless of their country of birth.

  1. Differences in Insomnia Symptoms between Immigrants and Non-Immigrants in Switzerland attributed to Emotional Distress: Analysis of the Swiss Health Survey. doi: 10.3390/ijerph16020289
  2. Sano, Y., Antabe, R., Kyeremeh, E., Kwon, E., Amoyaw, J. Immigration as a social determinant of troubled sleep in Canada: some evidence from the Canadian Community Health Survey-Mental Health Sleep Health 2019, 5, 135-140. DOI. 10.1016/j.sleh.2018.11.008
  3. Pascoe EA, Smart Richman L: Perceived discrimination and health: a meta-analytic review. Psychol Bull. 2009, 135: 531-554.
  4. Molarius, A., Granström, F., Feldman, I. et al.Can financial insecurity and condescending treatment explain the higher prevalence of poor self-rated health in women than in men? A population-based cross-sectional study in Sweden. Int J Equity Health11, 50 (2012). https://doi.org/10.1186/1475-9276-11-50

Round 2

Reviewer 1 Report

The authors have addressed the concerns from me and other reviewers and the revised manuscript appears much improved. Thanks for the efforts.

Reviewer 3 Report

Thank you for the authors tried to answer the questions and revised the manuscript appropriately. I have reviewed the author’s responses one by one including other reviewers. The authors have answered all the questions that were raised by me. Therefore, my recommendation now would be to accept.